# Mitigating IoT Privacy-Revealing Features by Time Series Data Transformation

**Feng Wang [1,*]**, **Yongning Tang [2]** and **Hongbing Fang [1]**

1   School of Engineering, Liberty University, Lynchburg, VA 24515, USA
2   School of Information Technology, Illinois State University, Normal, IL 61761, USA; ytang@ilstu.edu
*   Correspondence: fwang@liberty.edu

**Abstract:** As the Internet of Things (IoT) continues to expand, billions of IoT devices are now connected to the internet, producing vast quantities of data. Collecting and sharing this data has become crucial to improving IoT technologies and developing new applications. However, the publication of privacy-preserving IoT traffic data is exceedingly challenging due to the various privacy concerns surrounding users, IoT networks, and devices. In this paper, we propose a data transformation method aimed at safeguarding the privacy of IoT devices by transforming time series datasets. Based on our measurements, we have found that the transformed datasets retain the intrinsic value of the original IoT data and maintains data utility. This approach will enable non-expert data owners to better understand and evaluate the potential device-level privacy risks associated with their IoT data while simultaneously offering a reliable solution to mitigate their concerns about privacy violations.

**Keywords:** IoT; privacy leakage; time series; traffic pattern; data utility

## 1. Introduction

Collecting and sharing raw Internet of Things (IoT) packet-level network traffic data are expected to play a crucial role in improving existing IoT technologies, developing new applications, monitoring performance, and protecting privacy and security [1–6]. However, publishing such data is highly non-trivial since it may expose private and sensitive information not only about users but also about the IoT networks and devices from which it was generated.

Numerous studies have indicated that the patterns of packet-level network traffic data in the Internet of Things (IoT) differ significantly from those in traditional Internet traffic [7–9]. For instance, Figure 1 displays raw IoT packet-level data (including packet size and inter-arrival times (IAT)) captured from four surveillance cameras in the UNSW dataset [10]. These figures visually demonstrate that each IoT device displays its own *unique and periodic traffic patterns*.

Many methods have been proposed to identify and protect privacy-revealing features, which are specific characteristics or attributes of data that can be exploited to identify or infer sensitive information about an individual or a system. These features can range from identifiable personal information (IPI), such as names, addresses, or social security numbers, to subtler information, such as location data or browsing history. For instance, clustering packet traces from smart home IoT devices or using packet-size and IAT from wearable devices can help identify specific user activity [8,11].

However, most existing solutions focus on user-centric privacy-revealing features, such as those proposed in [12–18]. While these solutions prioritize user privacy, device-level privacy leakage in published IoT data can cause severe damage [19]. Traffic patterns in IoT packet-level data can reveal privacy information, allowing linkage attacks to IoT devices. For example, an attacker could exploit traffic rates to identify medical sensors attached to a

patient [19], if unique traffic patterns in raw IoT data are not obfuscated. This can lead to privacy attacks by adversaries, even if the published datasets are anonymized.

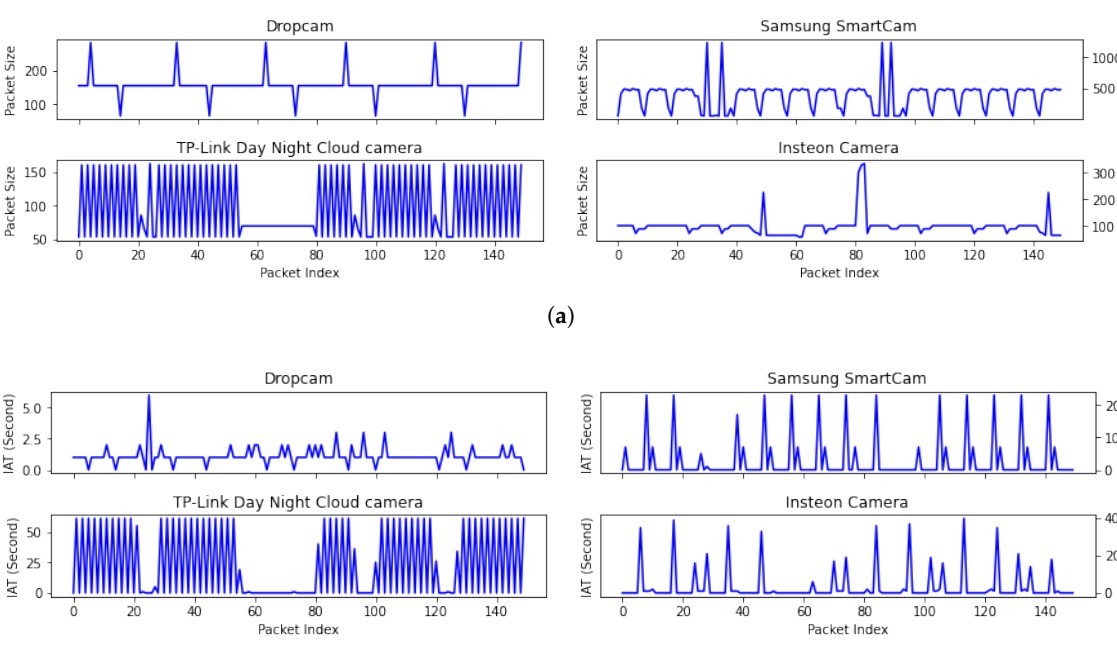

**Figure 1.** Unique and periodic packet size and IAT features of four surveillance cameras. (**a**) Packet size feature along with packet index; (**b**) IAT feature along with packet index.

This paper addresses the threat of an IoT membership inference attack (MIA), where an attacker uses a released IoT dataset to determine whether it was generated by a specific IoT device. This can lead to further security attacks by the attacker. Our goal is to protect the privacy of IoT devices in raw IoT packet datasets by preventing inference of device types, while preserving the original data's inherent value. Current anonymization techniques [12–18] for IoT traffic packet traces focus mainly on protecting user privacy, with little emphasis on device-level sensitive information.

This paper proposes a novel approach to protect device-level privacy in raw IoT packet data sharing. Our approach preserves device-level information while obfuscating sensitive patterns in the dataset, allowing data owners to mitigate the risk of privacy breaches in a cost-effective way. We use an efficient traffic reconstruction method to transform sensitive information while retaining useful data. Our results demonstrate that the proposed transformation method effectively obfuscates privacy-revealing patterns without compromising the value of the data. Machine learning models trained on the transformed datasets achieve similar accuracy to those trained on the original data, making the transformed datasets useful for data analytics tasks.

The main contributions of our study are as follows:

- We propose a novel method to protect device-level privacy in IoT data sharing by transforming time series datasets, preventing inference of IoT device types.
- We design an efficient traffic reconstruction method that preserves the value of the original data while protecting sensitive information. We evaluate the data utility of the transformed dataset using Euclidean distance and replicating studies, showing that our method effectively obfuscates privacy-revealing traffic patterns without sacrificing data utility.

The rest of the paper is organized as follows. Related works are discussed in Section 2. Section 3 introduces IoT device Membership Inference Attack (MIA). In Section 4, we elaborate the proposed transformation method. The data utility of transformed data is evaluated in Section 5. The paper is concluded in Section 6.

## 2. Related Works

Four main strategies have been proposed to reshape IoT traffic for privacy preservation: modification, replacement, perturbation, and synthesis.

The first strategy involves modifying the time stamps and feature values based on the time resolution and granularity of the feature values [20,21], but it can be costly.

In the replacement method, a field is mapped to a new value of the same type [16]. Truncation involves overwriting a field with fixed values [20], while generalization replaces specific data with a more general one through binning [22]. Finally, precision degradation removes the least significant information of a data field [21].

Obfuscating IoT time series data poses a significant challenge due to their unique characteristics. Traditional data obfuscation methods are often ineffective and inefficient for time series datasets. For example, the perturbation method [23,24] is commonly used to mask sensitive data by randomly adding crafted noises. However, controlling the perturbation level to preserve privacy without sacrificing the original data's value is a challenging task. To illustrate this challenge, we conducted an experiment where we added a significant amount of Laplace noise to two security camera datasets. The Laplace noise function is defined as $f(x, \mu, \lambda) = \frac{1}{\lambda} e^{-\left(\frac{|x-\mu|}{\lambda}\right)}$. We used different values of the parameter $\lambda$ to control the amount of generated noise. Figure 2 shows that when the parameter $\lambda$ changes from 6 to 46, the repetitive and consistent pattern of the Dropcam dataset is completely distorted, resulting in a degraded dataset utility. However, with the same amount of injected noise, the repetitive pattern of the Samsung SmartCam dataset is still recognizable. In general, achieving a balance between privacy level and model compatibility with perturbation methods is not satisfactory, as we will demonstrate in our experiments.

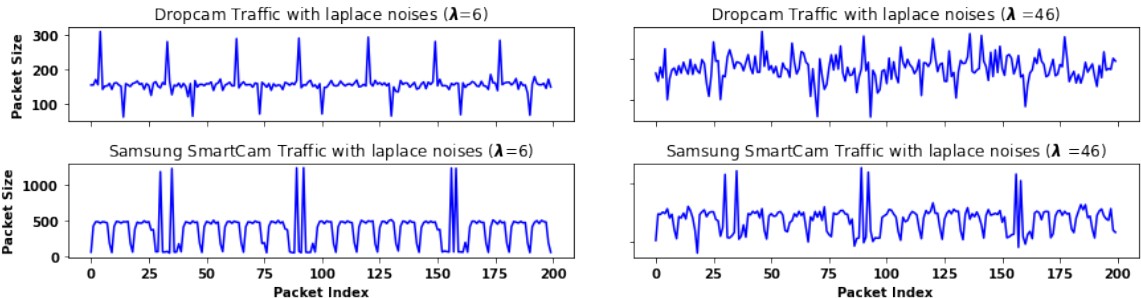

**Figure 2.** Packet size sequences from Dropcam and Samsung SmartCam are perturbed by adding varying amounts of Laplace noise.

Synthetic data [25–29] are another commonly used method to protect privacy. Recent synthetic methods focus on using Adversarial Network (GAN) models to construct synthetic IoT time series data [25,30–32]. Despite efforts to synthesize data, there is still a risk that the traffic patterns of the original data may be preserved, which could result in the disclosure of sensitive information from IoT devices. Generating time series datasets that preserve temporal dynamics using GAN is more difficult than generating tabular data. Concealing traffic patterns in IoT time series traffic data remains an unexplored research problem. These challenges have inspired us to develop a more effective approach to obfuscate device information in raw IoT packet data.

## 3. IoT Device Membership Inference Attack

In an IoT MIA, adversaries can identify IoT devices solely based on their traffic patterns disclosed from the released raw IoT packet data, without relying on device identifications such as IP or MAC addresses. Additionally, adversaries can leverage their own devices or use devices specified in the Manufacturer Usage Description (MUD) [10,33,34] to derive traffic patterns and associate them with specific IoT device types. Inferring information about IoT devices should not be confused with membership inferring attacks in machine learning [35], which determine if a record is in the model's training dataset.

We present two privacy attack scenarios resulting from the leakage of traffic patterns. In the first MIA scenario, an attacker seeks to determine if a Dropcam surveillance camera is present in a published dataset. A vulnerability in the Dropcam allows unauthorized users to halt its recording [36]. If an attacker can identify unique traffic patterns associated with home security cameras, the attacker can search for such a pattern in the published raw IoT packet data and launch a security attack if the targeted IoT device is successfully identified. Suppose the attacker uses a dataset collected on 6 October 2016, as the learning dataset. The attacker selects two snippets from the dataset and visualizes a sequence of packet size data, as shown in Figure 3a. By comparing the traffic patterns (two snippets in Figure 3b) from the published dataset (collected on 17 October 2018), the attacker can determine if a Dropcam is present in the dataset.

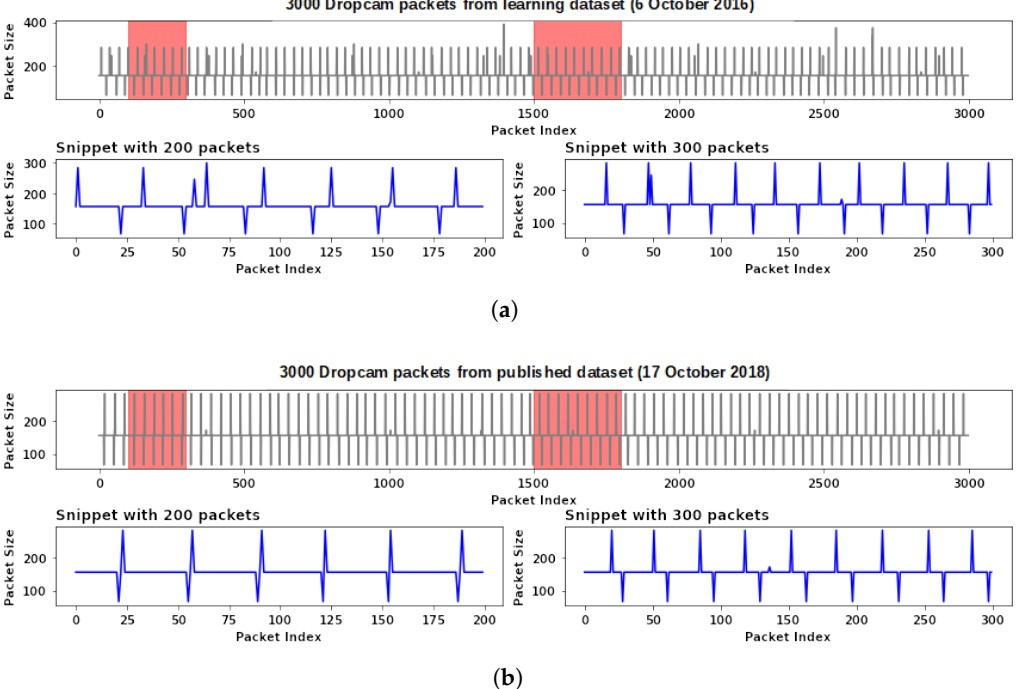

**Figure 3.** The visualization depicts a sequence of packet size data generated by a Dropcam in 2016 and 2018. The highlighted area shows the data points that were selected from datasets. (**a**) Packet size snippets from a learning dataset; (**b**) packet size snippets from a published dataset.

In the second MIA scenario, adversaries can identify if a Message Queuing Telemetry Transport (MQTT) broker is used in a published IoT dataset. A MQTT is an efficient publish/subscribe messaging transport protocol commonly used in IoT applications. A MQTT broker receives published IoT data, filters them by topics, and distributing them to corresponding subscribers. If adversaries can identify that a MQTT broker is used in a published dataset, they can launch a MQTT brute force attack [6].

## 4. Methodology

Data transformation aims to conceal the types of IoT devices by converting actual IoT time series data into a new dataset. This makes it impossible for an adversary to differentiate between the genuine device types. For example, as shown in Figure 4, the packet size feature of a Samsung SmartCam can be transformed into a new dataset with a similar feature of the Insteon camera. This makes it difficult for an attacker to identify whether a Samsung SmartCam is present in the published dataset.

### 4.1. Overview

To avoid confusion, we use the term "target dataset" to refer to the time series data that contains the desired feature. The packet-level pattern associated with this feature is referred to as the "target feature pattern" or simply the "target pattern". In Figure 4, the Insteon dataset is the target dataset, while its packet size feature is the target feature.

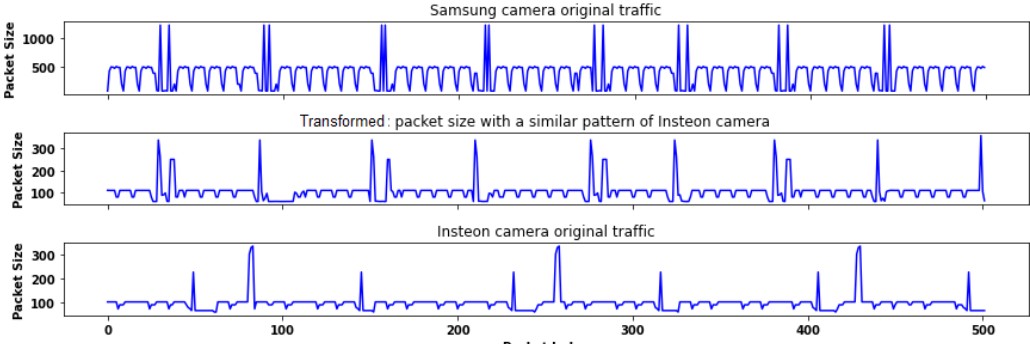

**Figure 4.** The transformed Samsung SmartCam dataset with a similar feature of the Insteon camera. (**Top**) the packet size feature of the original Samsung SmartCam data. (**Bottom**) the target feature pattern of Insteon camera data (target data). The middle figure shows the transformed data.

The architecture of our proposed data transformation method is shown in Figure 5. It consists of two primary components: transformers and a utility assessment. The transformers convert the input time series data into new data that contain either a desired feature pattern or a learned feature pattern. A data transformer selects a group of transformed candidates based on the similarity between the original and target data features. The purpose of the utility assessment is to maximize the selected feature similarity between the original and target datasets.

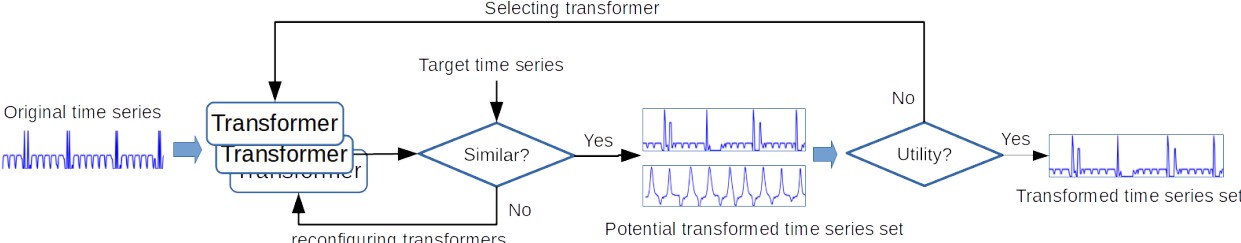

**Figure 5.** Architecture of data transformation. An original time series is the input of a set of transformers, and outputs are a set of transformed data candidates. A final transformed dataset is determined by the utility assessment unit.

To be more precise, the data transformation process involves two distinct phases. In Phase I, a group of transformers converts the input time series data into new datasets. Once Phase I is complete, a set of transformed candidates is generated, and Phase II begins. In Phase II, the transformed time series data are evaluated according to utility metrics, and the resulting similarity assessment is used to adjust the parameters used in training the transformers in Phase I.

### 4.2. LSTM-Based Transformer

The transformer is a powerful model that is capable of learning repeated patterns over a sequence of inputs. In particular, it is based on the Long Short-Term Memory (LSTM) model, which is a type of recurrent neural network (RNN) that can learn long-term dependencies in sequential data [37,38]. A LSTM network is composed of several key elements, including input gates, output gates, and forget gates, which allow it to selectively

remember or forget information over time. Additionally, LSTMs use a memory cell to store and retrieve information, which helps to overcome the vanishing gradient problem that is often encountered in traditional RNNs.

LSTM-based transformers are capable of transforming various time series data, including high-frequency repeated patterns and high-amplitude pulses. A set of LSTM models is trained using target datasets to learn target feature patterns. Since no labeled data are required to train the models, this method is unsupervised.

To facilitate readers' understanding of our proposed method, we present a summary of the symbols employed throughout the paper in Table 1. Formally, we consider a multivariate time series $X = x_t{}_{t=1}^T$ with $x_t \in \mathbb{R}^m$ being an $m$-dimensional observation (number of features). $T$ represents the total number of observations, and $t$ is the index of the measurements in time. Given a source time series data $X_s$, a transformer transforms $X_s$ to $X_t^*$ with a similar feature by learning a predictive function $\phi()$, i.e., $X_t^* = \phi(X_s)$.

**Table 1.** Table of symbols.

| Symbol | Description |
|---|---|
| $X$ | A multivariate time series |
| $x_{i,t}$ | The $i$-th feature value of time series $X$ at time $t$ |
| $T$ | The total number of observations in time series $X$ |
| $t$ | The index of the measurement in time |
| $X_s$ | Source time series data |
| $X_t^*$ | The transformed time series data with a similar feature to $X_s$ |
| $\phi()$ | The predictive function used to transform $X_s$ to $X_t^*$ |
| $x_{i,t_0:T}$ | The conditional distribution of a time series $X$ from time $t_0$ to $T$ for the $i$-th feature |
| $x_{i,1:t_0-1}$ | The past time series data for the $i$-th feature up to time $t_0 - 1$ |
| $t_0$ | The time point from which we assume $xi, t$ to be unknown at prediction time |

Specifically, each feature value of the time series $X$ is denoted by $x_{i,t}, i \in 1, \ldots, m$. The transformer's goal is to model the conditional distribution of a given time series $[x_{i,t_0}, x_{i,t_0+1}, \ldots, x_{i,T}]$, denoted as $x_{i,t_0:T}$. Given its past $[x_{i,1}, x_{i,t_0-2}, \ldots, x_{i,t_0-1}]$, denoted as $x_{i,1:t_0-1}$, the transformer represents the conditional distribution as: $P(x_{i,t_0:T}|x_{i,1:t_0-1})$. Here, $t_0$ denotes the time point from which we assume $x_{i,t}$ to be unknown at the prediction time.

When training transformers, we select multiple training instances by using sliding windows with different starting points $t_0$. Thus, the past $x_{i,1:t_0-1}$ and the prediction $x_{i,t_0:T}$ are with respect to the starting point of each training sample. The conditioning range size $t_0$ is a tunable parameter, while the prediction length $T$ determines the number of predicted data points. In this paper, we consider the situation where $T = 1$, or one prediction data point, to limit the number of predictions for each step and decrease the processing time.

### 4.3. Time Series Decomposition

To effectively process complex IoT traffic patterns, such as bursty traffic, we divide IoT traffic into two categories: active and inactive. Inactive traffic occurs when an IoT device is either in a sleep state, during which the device is not transmitting any data, or in a keep-alive state, where the device is transmitting routine periodic updates. Active traffic occurs when a non-periodic update event triggers the device, causing it to enter a bursty state where a larger amount of data is exchanged between the device and a server.

The traffic patterns of active and inactive traffic are distinct, with keep-alive traffic having a regular time pattern and small packets of constant size, while bursty traffic generates longer data bursts. In this paper, an IoT time series is decomposed into active and inactive datasets along with a timestamp. Time series data during active time period are defined as active data, while data during the inactive time period are called inactive data.

With the partition of active and inactive components, we introduce two LSTM-based transformers, denoted as $\phi_a()$ and $\phi_{in}()$. The original time series is decomposed as $X_t = X_a + X_{in}$, where $X_a$ and $X_{in}$ are the active and inactive datasets, respectively. The original time series is transformed into $X^*$, where $X^* = \phi_a(X_a, \theta_a) + \phi_{in}(X_{in}, \theta_b)$. The parameters $\theta_a$ and $\theta_b$ are used to train the active and inactive transformers, respectively. The conditioning range size $t_0$, the number of LSTM elements, and the batch size are the tunable parameters. The proposed heuristic method for the automatic decomposition of time series data into active and inactive datasets will be presented later.

### 4.4. Heuristic Method for Decomposition

We propose a heuristic approach for automatically partitioning time series data into active and inactive datasets. Our algorithm identifies the optimal partition of the original time series into two segments with distinct patterns. Specifically, we employ a heuristic search method to identify a threshold that minimizes the feature similarity between the active and inactive segments of a given time series. The active segment $Q$ and inactive segment $C$ are compared using the Euclidean distance metric:

$$p_d(Q, C) = \frac{1}{1 + \sqrt{\sum_{i=1}^{m}(q_i - c_i)^2}},$$

where, $m$ is the length of the segments and $q_i$ and $c_i$ represent the $i$-th elements of $Q$ and $C$, respectively. The similarity metric returns values between 0 and 1, where 0 indicates no similarity and 1 indicates maximum similarity.

Here is an example to illustrate the proposed approach. Suppose we have a time series dataset that contains information about the movement captured by a motion sensor placed in a building over the course of a week. We aim to partition the dataset into two segments: an "active" segment that contains motion sensor readings during periods of detected motion, and an "inactive" segment that contains motion sensor readings during periods of low or no activity.

To accomplish this, we use the proposed heuristic method to identify a threshold that separates the two segments. We calculate the feature similarity between the two resulting segments using the Euclidean distance metric. The closer the distance between the two segments, the less distinct their patterns are.

After several iterations, the algorithm identifies a threshold that yields two segments with significantly different patterns. The resulting "active" segment contains motion sensor readings when people are moving about the building, while the "inactive" segment contains motion sensor readings when the building is mostly unoccupied. The similarity metric is used to confirm that the two segments have distinct patterns, with a high dissimilarity score indicating a clear separation between the two segments.

### 4.5. Utility Assessment

The main challenge in data transformation is to strike a balance between preserving device privacy and maintaining the necessary utility of the original data. In this paper, we propose two methods to evaluate the utility of the transformed dataset.

The first method is based on replicating studies [39]. We perform anomaly detection on both the original and transformed data and draw conclusions. If the same conclusions are drawn from both datasets, then the transformed data are considered to have high utility.

The second utility assessment method is to evaluate the feature similarity between the transformed dataset and the original dataset. This similarity result is used to adjust the parameters of the LSTM transformers, such as the conditioning range size, number of LSTM elements, and batch size. The transformed dataset should have reduced but still have some feature patterns of the original dataset. By being similar to another device, the transformed dataset can confuse the adversary in determining its type.

The utility assessment unit determines which candidate maintains sufficient utility among a set of transformed candidates. If none of the candidates satisfy the requirement, the transformers need to be reconfigured or retrained to generate new transformed candidates.

After generating the transformed data, we perform post-processing by scaling the data to the range of the target dataset. The goal is to increase the semantic integrity of the transformed data to prevent adversaries from determining that the data are fabricated. For example, some transformed packet sizes may not exist in the target data. We use a feature mapping to learn the semantics from the target data, and process the transformed data.

Once the selected features of the original dataset are transformed, the transformed features and other unselected features will be saved in a cvs file. A low-level Python tool can generate pcap files from the cvs file.

### 4.6. Benchmark Datasets

In order to transform IoT raw data traces, our research requires benchmark datasets that meet two specific requirements. Firstly, the datasets must contain various types of IoT devices, which allows us to determine the traffic pattern of each individual device. Secondly, the datasets must provide raw traffic in pcap files, enabling our method to process raw packet traces into time series packet streams. Therefore, benchmark datasets that are flow based, such as the CIDDS datasets [40], are unsuitable for our purposes.

Our datasets are sourced from UNSW [10], and are the result of trace data that were captured over several years from a test-bed consisting of 28 distinct IoT devices. This data include two separate datasets, namely, BoT-IoT and UNSW-NB15. To train our LSTM models and assess the classification accuracy based on both the original and transformed data, we utilized the UNSW-NB15 dataset, which was captured over a six-month period. Additionally, we employed the BoT-IoT dataset to evaluate whether our transformed data still retained the critical features of IoT attacks, as outlined in our paper.

The UNSW-NB15 dataset comprises 100 GB of raw traffic in pcap files, with a total record count of two million. Conversely, the BoT-IoT pcap files are 69.3 GB and contain over 72,000,000 records. This dataset encompasses nine distinct types of IoT attacks, including Fuzzers, Analysis, Backdoors, DoS, Exploits, Generic, Reconnaissance, Shellcode, and Worms. In this study, we did not differentiate between each attack when evaluating data utility after transformation. Instead, we trained a binary classifier to detect these attacks, treating them as anomalies in our analysis.

## 5. Results

We have successfully developed a functional prototype of our proposed data transformation system, which utilizes LSTM-based transformers to transform time series datasets into new ones with selected features that closely resemble the target features. In order to evaluate the utility of the transformed datasets, we have employed two methods: feature similarity based on Euclidean distance, and replication of IoT classification and anomaly detection. Our measurements demonstrate that the proposed method is able to transform time series data while preserving the intrinsic values carried by the original data, including anomalies. Overall, our prototype provides strong evidence that our proposed method is effective and has the potential to be applied in real-world IoT scenarios.

### 5.1. Visualizing Transformed Data

Figure 5 illustrates the process we used to train transformers using UNSW datasets [10]. For each original dataset, we trained a set of transformers using different target datasets. Specifically, we trained four transformers using packet size and IAT features of four target cameras: Dropcam camera, Samsung SmartCam, TP-Link camera, and Insteon camera. We partitioned each target feature into an active and inactive set using fixed thresholds for the packet size. For example, we employ the decomposition method proposed in Section 4.4 to partition the Dropcam and Insteon datasets into active and inactive segments. The optimal thresholds for Dropcam and Insteon datasets are determined to be 216 and 120, respectively.

We then trained an active and an inactive transformer to learn the active and inactive parts of the selected target feature. Each transformer used an LSTM-based model with 50 units in the LSTM layer, one unit in the dense layer, and a batch size of 70. We developed the LSTM model using Python 3.7 based on TensorFlow and Keras. Finally, the trained target transformers transformed the active and inactive parts of the original data into new datasets with similar target features.

We generated new datasets for Dropcam by transforming its packet size and IAT features using the trained target transformers. The transformed features are presented in Figure 6 and Figure 7, respectively. These figures show the transformed Dropcam datasets that have similar packet size and IAT features to three other target cameras: Samsung SmartCam, TP-Link camera, and Insteon camera. From Figure 7, it is evident that the transformed features closely resemble the features of the target devices. We computed the Euclidean distance between the target feature and the transformed feature and found that the minimum distance was with the SmartCam camera. Therefore, we can transform the Dropcam dataset to a new dataset having features similar to the Samsung SmartCam.

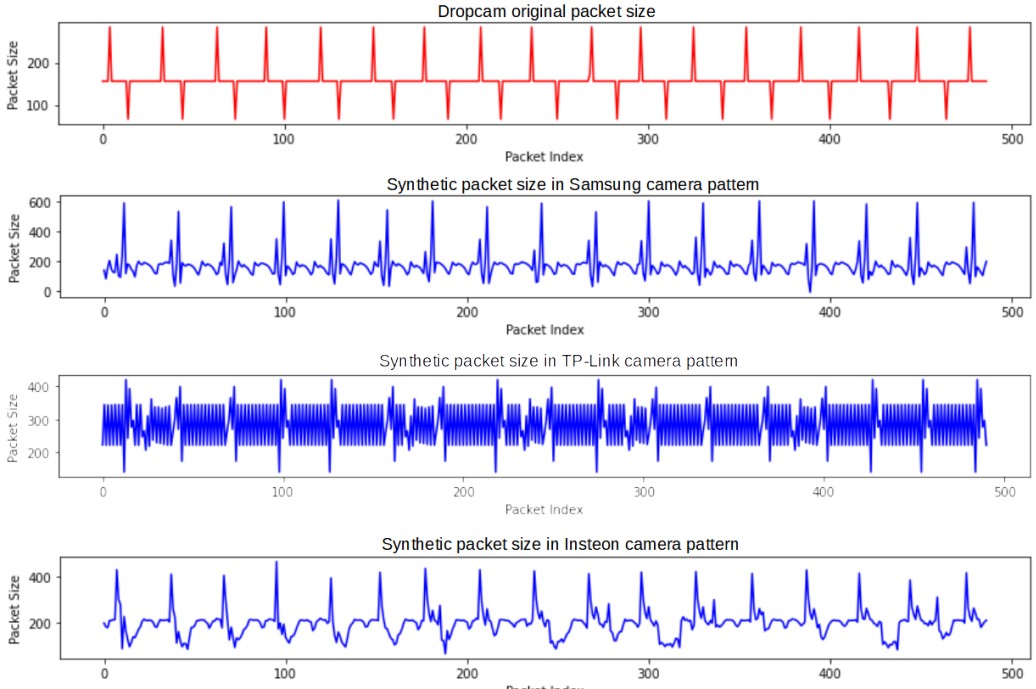

**Figure 6.** A sequence of transformed packet size of Dropcam with a similar feature of Samsung SmartCam, TP-Link, and Insteon cameras.

To showcase the effectiveness of transformers, we further demonstrate their ability to transform multiple original datasets using the same transformer. Specifically, we employed an Insteon camera's transformer to transform the packet size feature of Amazon Echo, LiFX smart bulb, and Belkin motion sensor datasets. The transformer used an LSTM-based model with 50 units in the LSTM layer, one unit in the dense layer, and a batch size of 70. The transformed packet size data are shown in Figure 8. The figure illustrates that the new datasets retain the target active and inactive patterns, making it difficult to identify the original features based on the transformed features. This result indicates that the transformer effectively captures and preserves the underlying patterns of the original datasets.

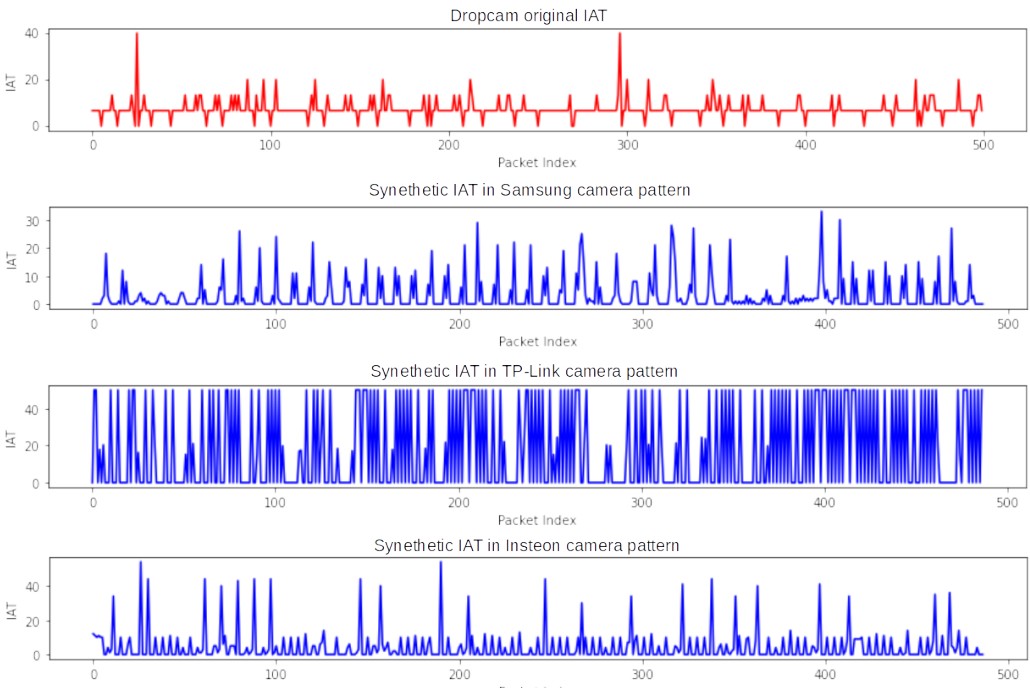

**Figure 7.** A sequence of transformed IAT of Dropcam with a similar feature of Samsung SmartCam, TP-Link and Insteon cameras.

### 5.2. Similarity Measurement

We begin by measuring the similarity between the transformed or perturbed data and the original data using a metric called similarity between the original data and transformed data (SOT). This similarity is computed by finding the minimum Euclidean distance between the two datasets. To evaluate the similarity, we select 100 time series points from both the original and transformed datasets using a sliding window and calculate the Euclidean distance between them. However, it is important to note that a transformed or perturbed feature with a minimum distance of zero can potentially leak the real feature. In Figure 9a, we present the SOT values of the perturbed method with two perturbation parameters and the transformation method. The figure shows that the perturbed time series data are still similar to the original data, while our transformation method produces a transformed feature that is different from the original feature. Later, we will evaluate whether the transformed data still retain the utility of the original data.

To evaluate the quality of the transformations, we also measure the similarity between the transformed data and the target data, which we refer to as similarity between the transformed data and the target data (STT). Similar to the SOT, we measure the STT using the Euclidean distance.

To select the best transformation candidates, we use both SOT and STT metrics. In Figure 9b, we plot $(x, y)$ pairs where $x$ represents the STT of the transformed data and $y$ represents the SOT. To choose the best transformation candidate, we look for the point closest to the identity line, which indicates that the transformed data are similar to both the target data and the original data. By selecting candidates close to the identity line, we ensure that the transformed data retain their original utility while also closely resembling the target data.

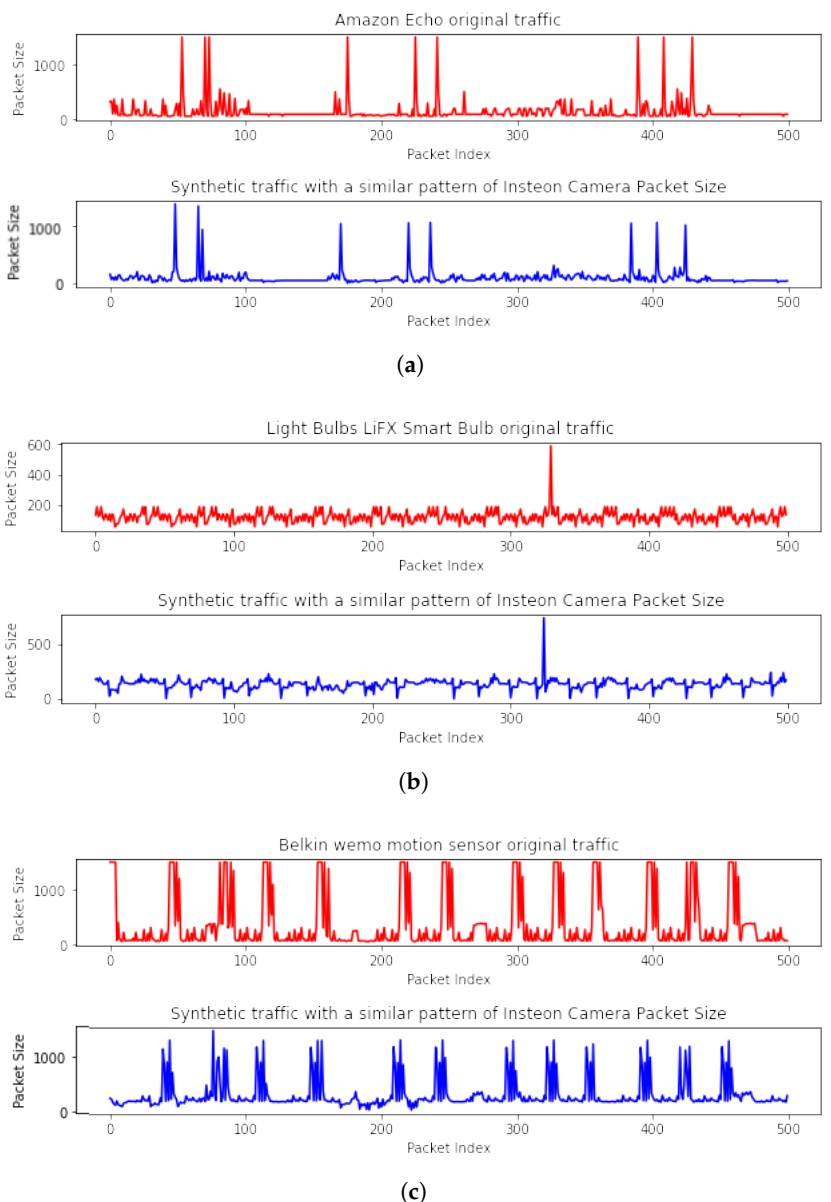

**Figure 8.** A sequence of packet size of three IoT transformed datasets with a similar packet size feature of Insteon camera. (**a**) Amazon Echo; (**b**) LiFX Smart Bulb; (**c**) Belkin wemo motion.

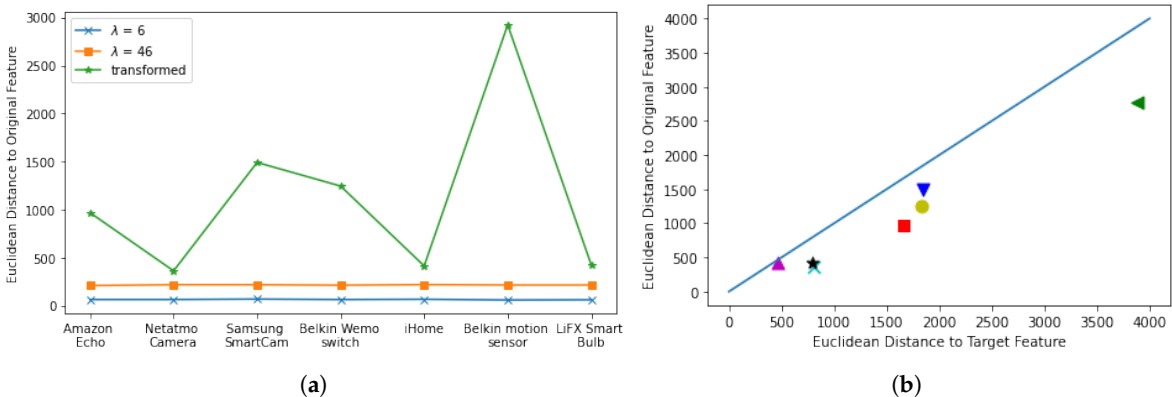

**Figure 9.** SOT and STT of packet size features of 7 IoT devices using a perturbation method and the proposed transformation method. (**a**) SOT; (**b**) SOT and STT of 7 IoT devices, each represented by a different shape.

We evaluate the utility of our approach by training a machine learning model to classify the transformed and perturbed data ($\lambda = 46$) and the original data. We combine the transformed/perturbed data with the original data and randomly split the dataset into 70% training data and 30% evaluation data. We train an XGBoost model on the training data and evaluate it on the evaluation data. The results are shown in Figure 10. The model is unable to distinguish between the perturbed data and the original data, indicating that the perturbed data are still similar to the original data. However, the model can accurately distinguish between the transformed data and the original data, indicating that our transformation approach effectively alters the data while preserving their utility.

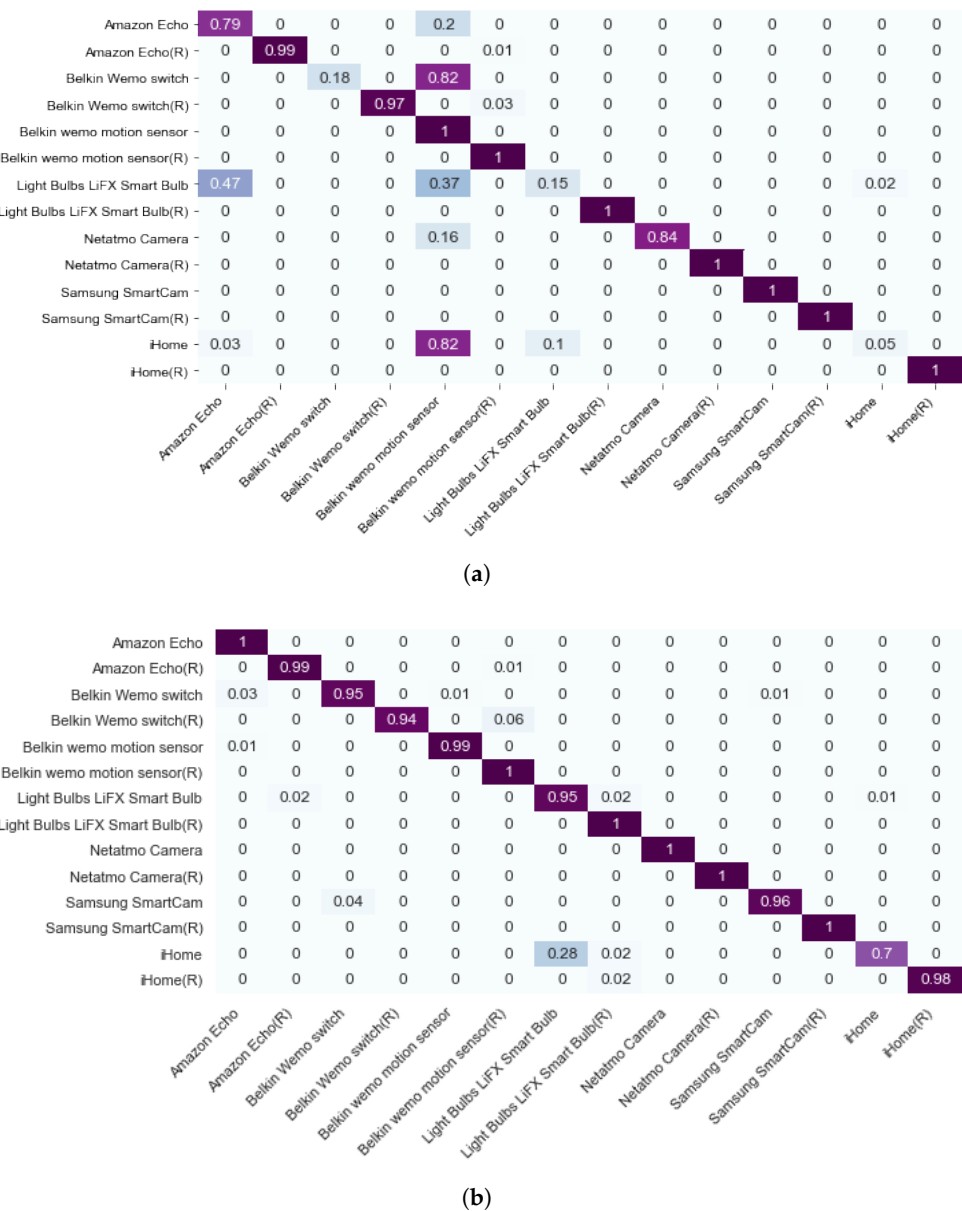

(a)

(b)

**Figure 10.** Confusion matrix of classification results of the original, perturbed and transformed data, with dark purple representing correctly classified instances and light blue representing incorrectly classified instances. (**a**) The original and perturbed data; (**b**) The original and transformed data.

Here, we want to clarify that it is challenging for an attacker to accurately identify sensitive IoT devices using transformed datasets. Even if we obfuscate the traffic features of a device, such as changing the Dropcam data to Insteon data, an attacker may still be able to classify the sensitive devices from the rest of the devices using the transformed

data, for example, as shown in Figure 10. However, the attacker cannot identify the device because the transformed data do not resemble the original data, but instead have a higher likelihood of resembling the target data. Thus, the transformed data do not contain any privacy-revealing features.

### 5.3. Utility Measurement

To assess the utility of the transformed data, we conducted IoT classification and anomaly detection on both the original and transformed datasets. Firstly, we evaluated the IoT classification results by classifying seven IoT devices using both the original and transformed data. We trained a model using the original data as the baseline measurement and achieved a classification accuracy that was not 100% due to the use of a single feature. However, we found that the perturbation method had a negative effect on the classification accuracy, especially when using a large $\lambda$ value, as shown in Figure 11b,c. In contrast, the transformation method had a minimal impact on accuracy, as seen in Figure 11d, where we were able to achieve almost the same classification results using the transformed data. Following this, we visualized the transformed anomaly and presented the anomaly detection results.

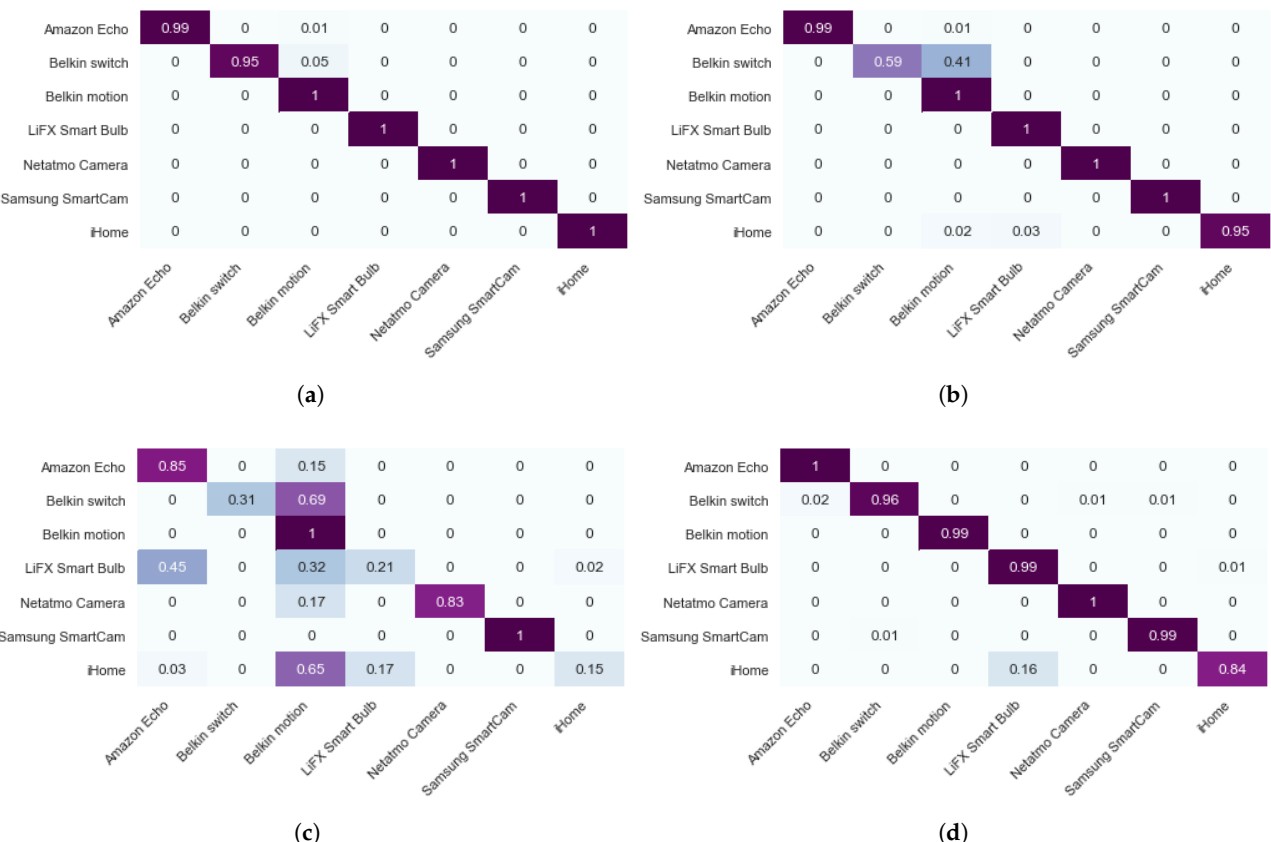

**Figure 11.** Confusion matrix of classification results of seven IoT devices based on packet size feature, with dark purple representing correctly classified instances and light blue representing incorrectly classified instances. (**a**) Original data; (**b**) Perturbed data ($\lambda = 6$); (**c**) Perturbed data ($\lambda = 46$); (**d**) Transformed data.

As shown in Figure 12, the anomalies in the Samsung SmartCam original dataset are transformed into the new data, which are still noticeable. Two malicious attacks in UNSW data impact a Belkin switch and a motion sensor, as shown in Figure 13. After transforming the real data, we find the new data preserve the anomalies occurring in the original time series data. We use the transformed data, which contain anomalies, and the original data to train a new model to detect the anomaly records. It is important to note that we did

not differentiate each attack. A binary classifier is trained to detect normal or attack traffic. We compare the accuracy of anomaly detection based on perturbed and transformed data. In Figure 14a, a base-line anomaly detection result shows that using the packet size feature cannot detect all anomalies. The model trained by the perturbed data cannot detect most of the anomalies. On the contrary, the model trained by the transformed data has a little higher accuracy than using the original data.

To summarize, the transformation method does not impact data utility. The proposed transformation method not only transforms traffic patterns, but also keeps feature characteristics of the original data. As a result, the transformed data are deemed to have high utility. The synthetic data can still be used to understand data characteristics and main variables needed for data mining.

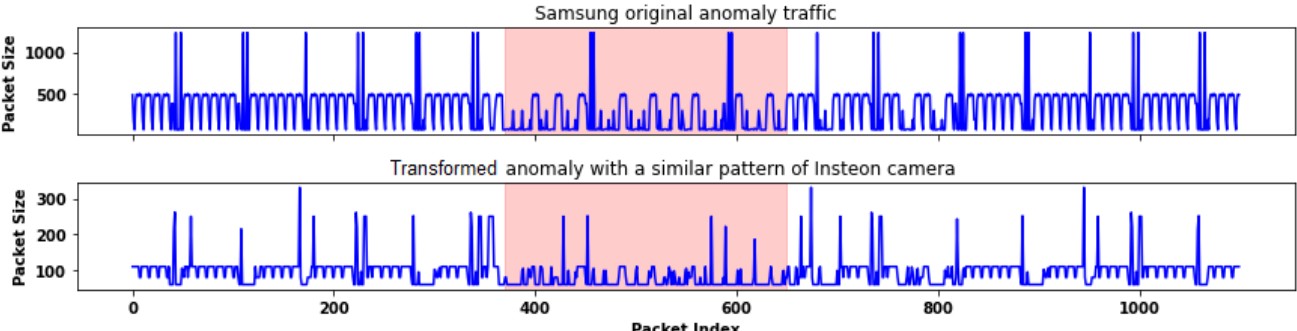

**Figure 12.** Reconstructed anomalies of transformed Samsung SmartCam data with an Insteon-like packet size feature. The anomalies are highlighted.

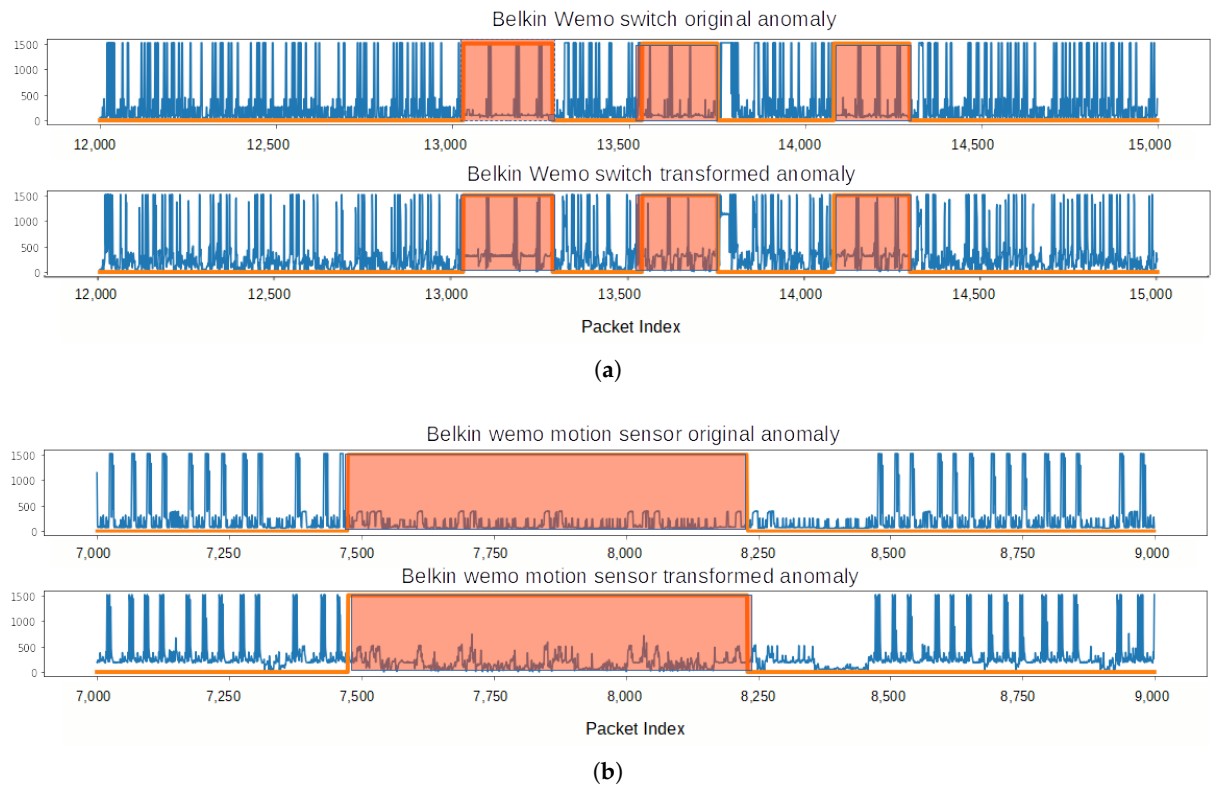

**Figure 13.** Reconstructed anomalies of Belkin data with an Insteon-like packet size feature. The anomalies are highlighted. (**a**) Belkin Switch; (**b**) Belkin wemo motion.

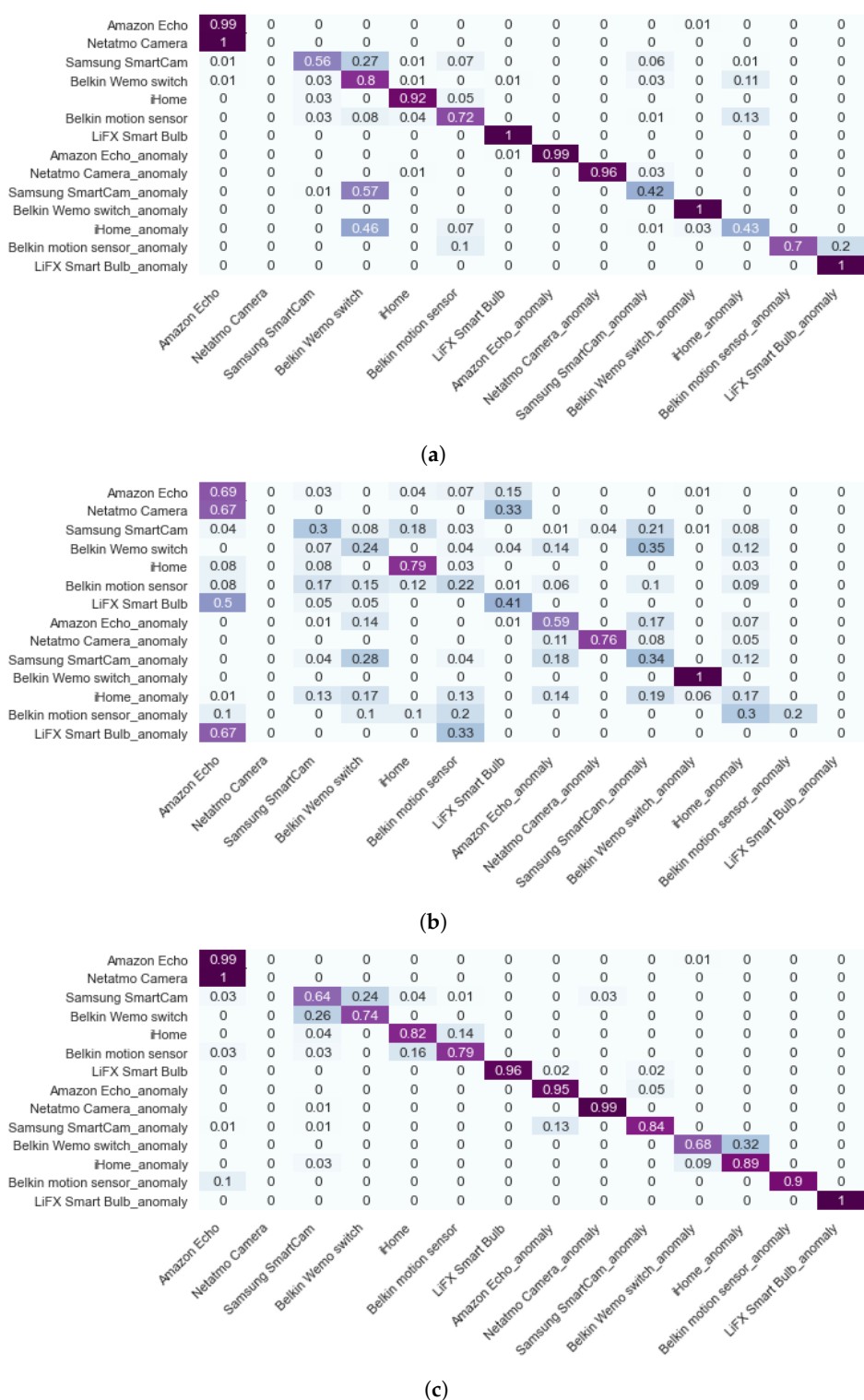

**Figure 14.** Confusion matrix of anomaly detection using perturbed and transformed data, with dark purple representing correctly detection instances and light blue representing incorrectly detection instances. (**a**) Raw data; (**b**) Perturbed data; (**c**) Transformed data.

## 6. Conclusions and Future Work

The findings of this study highlight the potential risks that IoT device membership inference attacks (MIA) pose to the privacy of IoT devices. While there has been some research on protecting privacy in IoT systems, there is still a lack of focus on protecting

device-level privacy-revealing features. As a result, this work proposes a novel method to safeguard the privacy of IoT devices by preventing adversaries from inferring their types.

One of the main challenges in developing a privacy protection method for IoT devices is to find a balance between preserving the inherent value of the original IoT data while restricting what the attacker can learn. This work has addressed this challenging problem by proposing a time series data transformation method that effectively reshapes IoT packet data while preserving the important features and ensuring that the transformed dataset still contains useful information.

The proposed time series data transformation method appears to be a cost-effective and efficient solution to mitigate device-level vulnerability. The results of the study demonstrate that the transformed dataset retains the intrinsic value of the original IoT data while preserving data utility. This means that the transformed dataset can still be used for various applications, such as anomaly detection, without revealing sensitive information about the device type.

Overall, the proposed solution is an important step towards safeguarding the privacy of IoT devices from membership inference attacks. Further research can be conducted to evaluate the effectiveness of the proposed method in different IoT environments and to optimize the method for larger-scale IoT systems. We plan to incorporate the TON_IoT datasets [41] as part of our evaluation of the data utility resulting from our transformation method. These datasets are obtained from UNSW and comprise both Industry 4.0/Internet of Things (IoT), and Industrial IoT (IIoT) datasets, providing us with an opportunity to further assess the effectiveness of our transformation method.

While the proposed data transformation method appears to be effective in mitigating device-level vulnerability in IoT systems, there is still a need to explore other transformation methods. One of the limitations of the LSTM-based transformer used in this study is that it operates as a blackbox, which means that the granularity of changing privacy-revealing features is not controllable. This lack of control over the transformation process could potentially lead to the loss of important information or features that could be useful for other applications.

To address this limitation, future research can explore the use of other time series transformers that provide more control over the transformation process. One area for further exploration is the use of time series transformers [42] to convert time series data. Time series data are a critical component of IoT systems, and the use of time series transformers could offer a more effective way to preserve the privacy of IoT devices. Time series transformers are specifically designed to handle time series data, which could make them more effective at preserving the important features of IoT data while preventing adversaries from inferring device types.

In conclusion, while the proposed data transformation method is an important step towards protecting the privacy of IoT devices, further research is needed to explore other transformation methods that provide more control over the transformation process. The use of time series transformers is a promising area for future exploration, as it could offer a more effective way to preserve the privacy of IoT devices, especially when dealing with time series data.

**Author Contributions:** Conceptualization, F.W. and Y.T.; methodology, F.W. and Y.T.; software, F.W.; validation, F.W., Y.T. and H.F.; data curation, F.W.; writing—original draft preparation, F.W.; writing—review and editing, Y.T. and H.F. All authors have read and agreed to the published version of the manuscript.

**Funding:** This research received no external funding.

**Data Availability Statement:** The data presented in this study are openly available in UNSW IoT Traces at https://iotanalytics.unsw.edu.au/iottraces.html, and UNSW attack data at https://iotanalytics.unsw.edu.au/attack-data.html.

**Conflicts of Interest:** The authors declare no conflict of interest.

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
