# Peer review of "Mitigating IoT Privacy-Revealing Features by Time Series Data Transformation"

_jcp, doi:10.3390/jcp3020012_

Round 1

Reviewer 1 Report

Although I find the reduction method proposed by using time series on IoT networks, especially on privacy in the study, successful in general, I believe that working with the following regulations may be more valuable.

-While the contribution of the study to the literature is clearly explained in the introduction, the reason why such a study is needed (motivation) and the limitations of other studies in the literature are not mentioned. For this reason, the reorganization of the entrance part will make the study more valuable.

-In order to measure the effectiveness of the proposed method, it is useful to evaluate it in different IoT datasets (CIDDS, ToN-IoT etc) and under different IoT attacks (phishing, sinkhole, malware, DoS/DDoS etc).

Author Response

Thank you for your constructive comments and suggestions. We have carefully incorporated them in the revised paper. 

Reviewer 2 Report

This paper proposes a way to modify/transform the packet sizes in IoT network traffic to protect against identification of IoT devices, while at the same time preserving utility of the traffic dataset. This is not a new problem, but the proposed approach is new to my knowledge (based on LSTM transformers).

Detailed comments:

1. p. 2, line 53 - What are IAT features?

2. p. 6, line 175 - What are LSTM elements? They have not been defined before in this paper.

3. p. 6, section 3.3 - The decomposition method needs more details. Please describe how it takes an input time series X and produces a partition of X into two time series Q and C.

4. p. 7, lines 221-222 - 'For example, we use packet size 216 and 120 to split Dropcam and Insteon data, respectively.' Please describe in more detail how this is done.

5. p. 7, section 4 - Please give a brief description of the dataset you use, e.g. how many IoT devices, what sort of traffic is captured for these devices, how many packets, how long (in time) is the capture.

6. In what way is privacy being protected? Judging from the IoT classification results (Figs 10 and 11) it seems it is possible for the attacker to correctly identify which IoT device has generated the traffic data. Doesn't that mean that there is not much privacy?

7. Does the proposed protection method apply to other features? Or can it only be applied to packet sizes?

Author Response

Thanks for your comments and suggestions. We have carefully incorporated them in the revised paper. 

Reviewer 3 Report

1-The summary did not explain the method of work clearly and simplified. Also, it was not explained how to demonstrate the efficiency of the system by clarifying the approved standards or their ratios

2-The first letter of each keyword is written with a capital letter. It is also preferable to choose five keywords instead of three

3-In this sentence, IOT is mentioned many times: Collecting and sharing raw Internet of Things (IoT) packet-level network traffic data are expected to play a crucial role in improving existing IoT technologies, developing new IoT applications, monitoring IoT application performance, and protecting IoT privacy and security [1–6].

4-Figure 1 is not mentioned in the research paper

5-What are the contributions made by the researcher compared to previous research papers? This has not been clarified

6-The term MQTT when first mentioned did not indicate its abbreviation

7-Unification of the mechanism for writing titles for sections according to the permitted style of the journal in terms of writing the first capital letter for the first word only or a capital letter for each word of the title

8-It is preferable to add a table of the symbols mentioned in the research paper

9-The related work section is placed at the front of the paper, not at the end

10-Rewrite the discussion better to clarify the conclusions reached by the researcher and the future work of those working in this field

11-Figure 14 is placed in its proper place, not in the references

12-References are written in a uniform style

Author Response

Thanks very much for this insightful comment. We sincerely appreciate your time in reading the paper, and our point-to-point responses to your comments are given in the attached file. 

Round 2

Reviewer 2 Report

The authors have addressed my comments. Happy to recommend acceptance.